# The Potential Roles of Ferroptosis in Pathophysiology and Treatment of Musculoskeletal Diseases—Opportunities, Challenges, and Perspectives

**DOI:** 10.3390/jcm12062125

**Published:** 2023-03-08

**Authors:** Yunxiang Hu, Yufei Wang, Sanmao Liu, Hong Wang

**Affiliations:** 1Department of Orthopedics, Dalian Municipal Central Hospital Affiliated of Dalian Medical University, No. 826, Southwestern Road, Shahekou District, Dalian 116021, China; 2School of Graduates, Dalian Medical University, No. 9, West Section of South Lvshun Road, Dalian 116044, China; 3Department of Anesthesiology, The Second Affiliated Hospital of Dalian Medical University, Dalian 110623, China

**Keywords:** ferroptosis, osteoporosis, osteoarthritis, osteosarcoma, intervertebral disc degeneration, spinal cord injury

## Abstract

Ferroptosis is different from other forms of cell death, such as apoptosis, autophagy, pyroptosis, and cuproptosis, mainly involving iron metabolism and lipid peroxidation. Ferroptosis plays an important role in various disease, such as malignant tumors, neuron-degenerative diseases, and cardiovascular diseases, and has become the focus of current research. Both iron overload and lipid peroxide accumulation contribute to the occurrence, development, and treatment of musculoskeletal diseases, such as osteoporosis, osteoarthritis, osteosarcoma, intervertebral disc degeneration, and spinal cord injury. For a better understanding of the potential roles ferroptosis may play in pathophysiology and treatment of common musculoskeletal disorders, this article briefly reviewed the relationship and possible mechanisms. Through an investigation of ferroptosis’ role in musculoskeletal diseases’ occurrence, development, and treatment, ferroptosis could offer new opportunities for clinical diagnosis and treatment.

## 1. Introduction

Cell death can appear in physiological and pathological processes such as embryonic development and biological internal environment balance. Cell death patterns are divided into two major categories: non-programmed cell death and programmed cell death [1]. Non-programmed cell death is passive death, mainly including autolysis or heterolysis; programmed cell death is active death, including autophagy, apoptosis, programmed necrosis, ferroptosis, pyroptosis, and cuproptosis [2,3,4]. Studies have shown that a number of diseases are closely associated with ferroptosis in terms of physiology and pathology [5], ischemia-reperfusion injury [6], neurological diseases [7], and bone metabolism diseases [8]. Ferroptosis is a non-apoptotic mode of cell death, which is significantly different from other programmed cell deaths (including traditional apoptosis, autophagy, necrosis, pyroptosis, etc.), and is mainly characterized by the accumulation of iron-dependent reactive oxygen species (ROS) in cells, thus promoting cell death. Ferroptosis refers to the catalysis of unsaturated fatty acids highly expressed in cell membranes under the action of Fe^2+^ or lipoxygenase, which undergo liposome peroxidation and ultimately lead to cell death [9]. Musculoskeletal disorders such as osteoporosis, osteoarthritis, osteosarcoma, intervertebral disc degeneration, and spinal cord injury are closely related to iron overload and lipid overload. For a better understanding of the potential roles ferroptosis may play in pathophysiology and treatment of common musculoskeletal disorders, this article briefly reviewed the relationship and possible mechanisms. 

## 2. Materials and Methods

In our study, published peer-reviewed journal papers in English were included; PubMed, Google Scholar, and Web of Science databases were searched from inception until October 2022. Keywords used for searching included ferroptosis and osteoporosis, or osteoarthritis, osteosarcoma, intervertebral disc degeneration, or spinal cord injury. Following that, manual searches of the reference lists of the studies found in the databases were conducted. Meta-analyses and review articles were excluded as research sources. All articles (title and abstract) were screened by three authors using the predefined eligibility criteria (original studies focusing on ferroptosis and the mentioned musculoskeletal diseases). If two of the three authors agreed that the manuscript met the eligibility criteria, it was considered suitable for further analysis (primary literature).

### 2.1. Overview and Main Characteristics of Ferroptosis

In 2003, Dolma et al. [10] found that erastin had the ability to induce cancer cell death, and typical features such as caspase 3 activation and nuclear morphological changes (nuclear fragmentation, chromatin condensation, and cell volume reduction) did not appear during erastin-induced cell death, while caspase inhibitors also had no significant inhibitory effect on erastin-induced cell death, so it was speculated that there may be a brand-new mode of cell death that is different from the classical mode of apoptosis. In 2008, Yang and Stockwell [11] found that Ras-selective lethal compounds shared similarity with erastin and detected that both elevated ROS and iron levels were strongly associated with Ras signaling. Until 2012, Dixon et al. [12] demonstrated that erastin-induced cell death differed from necrosis, apoptosis, and autophagy in morphology, biochemical properties, and genetics, characterized by massive intracellular accumulation of iron-dependent ROS, and based on this feature it was officially named as ferroptosis. As a novel mode of nonapoptotic cell death, ferroptosis differs from other programmed cell deaths (e.g., apoptosis, autophagy, necrosis, or pyroptosis) [2,9,13,14,15] (Table 1). 

### 2.2. Mechanism of Ferroptosis 

Ferroptosis is iron-dependent and differs from novel modes of programmed cell death such as apoptosis, necroptosis, and autophagy [16]. The occurrence of ferroptosis involves the abnormal metabolism of iron, accumulation of reactive oxygen species, and metabolism of lipids. In iron metabolism, iron, as an important trace element, participates in a number of important physiological activities such as oxygen transport and hemoglobin synthesis. Iron excess in the body is caused by factors such as excessive iron uptake, genetic mutations, and aging, because the body lacks an effective method of eliminating iron [17]. Pathologic conditions in many diseases are associated with iron overload or metabolic abnormalities, such as overload of iron ions that increases the risk of joint disease [18], neurodegenerative diseases, etc. [19]. It has been confirmed that the level of iron ions increases significantly in cells undergoing ferroptosis, so abnormal accumulation of iron ions is considered to be one of the important markers of ferroptosis. Under normal physiological conditions, iron exists in the form of Fe^3+^ and Fe^2+^, and Fe^2+^ is oxidized to Fe^3+^ in vivo, which in turn forms a complex with transferrin and is reduced to Fe^2+^ under the action of ferric reductase to maintain iron homeostasis. Excess iron can generate ROS accumulation through the Fenton or Haber–Weiss reaction, initiate lipid peroxidation, and induce the development of ferroptosis [20]. Abnormal accumulation of lipid peroxidation ROS is one of the important factors inducing ferroptosis. Lipid peroxidation refers to the carbon-carbon double bond process in which oxidants attack lipids such as polyunsaturated fatty acid (PUFA). PUFAs have been reported to be highly sensitive to lipid peroxidation and are considered a prerequisite for the development of ferroptosis [21]. Lipid metabolism-related genes lysophosphatidylcholine acyltransferase 3 (*LPCAT3*) and acyl coenzyme A synthase long chain member 4 (*ACSL4*) are indispensable and important genes for the inhibition of ferroptosis [22], whereas arachidonic acid (AA) is regarded as the most frequently consumed PUFA in ferroptosis [23]. Abnormal accumulation of ROS leads to GSH depletion and reduced GPX4 activity [24], while GPX4 is currently the only known glutathione peroxidase (GPX) for liposomal peroxide reduction and plays a key role in the development of ferroptosis. Multiple compounds induce ferroptosis, and their signaling pathways vary, but the end result is a decrease in GPX activity and an increase in lipid peroxidation, leading to the development of ferroptosis, so GPX4 is considered to be a key regulator of ferroptosis [25]. At present, there are two main mechanisms of ferroptosis, which are inhibition of GPX4 and inhibition of cystine/glutamate reverse transport receptor (systemXc) activity. Both RSL3 and erastin are ferroptosis inducers, of which RSL3 is a GPX4 inhibitor that directly inhibits GPX4 to inactivate it, resulting in massive accumulation of ROS and ultimately inducing ferroptosis [26]. Various studies have confirmed that key mediators of ferroptosis are iron metabolism and lipid peroxidation (Figure 1).

## 3. Osteoporosis and Ferroptosis

Osteoporosis is a common systemic skeletal disorder that increases bone fragility and fracture risk. Its pathogenesis is often associated with hormonal imbalance, nutritional factors, and genetic factors. Significant features of ferroptosis include iron overload and increased ROS, both of which can influence the progression of osteoporosis. 

### 3.1. Iron Overload

It has been confirmed that iron overload is an important risk factor for the development of osteoporosis [27] and is positively correlated with the development of osteoporosis [28]. Meanwhile, abnormal accumulation of iron inhibits the activity of osteoblasts and increases the activity of osteoclasts [29]. When there is an excess of cellular iron, bone formation and bone destruction are disrupted, destroying bone balance, and bone weakening is one of the common features of iron overload, which is manifested as osteopenia, bone microarchitecture and biomechanical changes, and frequent fractures [30]. Iron overload can inhibit the activity of osteoblasts to a certain extent, affect their differentiation and mineralization process, initiate the activation and differentiation of osteoclasts, cause bone loss, and ultimately lead to osteoporosis [31,32]. Osteoporosis induced by iron overload has been studied in animal models and clinical practice. Chronic administration of iron dextran results in tissue iron overload and osteoporosis in mice [33]. Studies have shown that in addition to estrogen deficiency, postmenopausal women also show signs of iron overload, which may be a key factor predisposing elderly women to osteoporosis; iron chelators can effectively inhibit iron accumulation and osteoclast differentiation, reduce bone destruction, improve bone tissue microarchitecture, and prevent bone loss [34]. Iron overload can affect the balance of bone metabolism and induce the occurrence of osteoporosis. Controlling iron overload can maintain the balance of iron homeostasis to a certain extent, which may be one of the effective means to prevent osteoporosis. 

### 3.2. ROS

In studies, iron ions have been found to stimulate the differentiation of osteoclasts and the resorption of bone by producing ROS [35]. Sithole et al. [36] reported that G protein-coupled receptor 120 signaling may inhibit osteoclast formation and resorption by downregulating ROS levels in RAW264.7 murine macrophages, thereby delaying the development of osteoporosis. He et al. [37] used rhaponticin, an extract of rhubarb, to intervene with RANKL-induced osteoclasts in experiments, which can cause a decrease in the production and increased consumption of ROS, and then increase antioxidant activity, ultimately inhibiting osteoclastogenesis and resorption, and can also provide potential therapeutic drugs for osteoporosis prevention and treatment. 

### 3.3. Possible Mechanisms in Osteoclasts and Osteoblasts

Under normoxia, ferritophagy and iron starvation responses confirm ferroptosis’ involvement in osteoclast differentiation; as a result of stimulation with RANKL, the expression of *MDA* and prostaglandin endoperoxide synthase 2 (*PTGS2*) genes in bone marrow-derived macrophages (BMDMs) increased, a decrease in GSH and iron levels in the culture medium supernatant was observed, as well as an accumulation of iron in mitochondria [38]. Qu et al. [39] found that zoledronic acid induces osteoclast ferroptosis through ubiquitination and degradation of p53 via FBXO9. Based on a glucocorticoid-induced osteoporosis (GIOP) model with high doses of dexamethasone, Lu et al. [40] reported that high-dose dexamethasone (10 μM) can induce osteoblast ferroptosis in mice, and a downregulation of GPX4 and system xc- may be possible. Interestingly, Balogh et al. [41] found that iron inhibited osteogenic differentiation of MSCs, and iron overload in mice is associated with increased ferritin levels and decreased *RUNX2* in compact bone osteoprogenitor cells (Figure 2).

## 4. Osteoarthritis and Ferroptosis

Osteoarthritis (OA) is a chronic degenerative disease caused by a variety of factors, and knee osteoarthritis has become the fourth leading cause of disability in the world [42]. An increasing number of studies have found that iron overload is associated with OA, and abnormal iron metabolism is considered to be one of the risk factors for OA development [8]. The concentration of iron in synovial fluid was significantly higher in OA patients than in healthy subjects [43], and the degree of joint injury was positively correlated with the increase of serum ferritin [44]. By establishing a mouse model of OA induced by iron overload and surgery, it was found that iron overload caused high expression of adisinte grinandmetalloproteinasewiththrombospondinmotifs5 (ADAMTS5) and matrixmetal lopmteinase13 (*MMP13*); meanwhile, divalent metal ion transporter 1 (*DMT1*) played a key role in the progression of OA induced by iron overload [45]. Intra-articular injection of ferrostatin 1 into a mouse model of OA upregulated the expression of type II collagen and delayed OA development [46]. In addition, some scholars have found that D-mannose can protect cartilage and reduce the progression of OA by attenuating the sensitivity of chondrocytes to ferroptosis [47]. Both interleukin 1β (IL 1β) and ferric ammonium citrate (FAC) induced abnormal accumulation of ROS in articular cartilage, leading to chondrocyte death, while erastin, a ferroptosis inducer, upregulated the expression of *MMP13* in cartilage and downregulated the expression of type II collagen in chondrocytes. Synovitis is an important pathological process of OA and has been confirmed to be associated with ferroptosis. Some scholars have used lipopolysaccharide to treat synoviocytes to construct synovitis cell models and found that iron content and *MDA* in synoviocytes are increased; solute carrier family member 3 member 2 (*SLC3A2L*), solute carrier family member 7 member 11 (*SLC7A11*), and GPX4 levels are decreased, while icariin can reduce iron content and inhibit ferroptosis [48]. There is currently evidence to suggest that abnormal intracellular iron metabolism causes a large amount of iron accumulation involved in the pathological process of synovial changes, cartilage degeneration, and subchondral bone remodeling. Synovitis leads to synovial hyperplasia and angiogenesis and also promotes chondrocyte ferroptosis, in addition to increased proliferation and activity of osteoclasts and inhibition of osteoblast function and proliferation, leading to subchondral bone remodeling [49] (Figure 3).

## 5. Osteosarcoma and Ferroptosis

Among children and adolescents, osteosarcoma is the most commonly occurring malignant bone tumor. The main treatment methods are chemotherapy and surgery. Ferroptosis has been found to play a key role in immune function and tumor development. β-Phenylethylisothiocyanate (PEITC) has anticancer potential. Lv et al. [50] concluded that PEITC was found to inhibit K7M2 bone marrow osteosarcoma cells, induce oxidative stress, GSH inactivation, increase *MDA* and ROS, and decrease GPX4, and it was considered that PEITC activated the MAPK signaling pathway, and ROS generation was the main cause of PEITC-induced cell death. Shi et al. [51] found that Tirapazamine (TPZ) could effectively inhibit the proliferation and metastasis of osteosarcoma. By measuring the level of related ferritin in osteosarcoma cells, it was confirmed that TPZ induced a significant decrease in the expression of key ferroptosis proteins such as *SLC7A11* and GPX4, an increase in *MDA* and Fe^2+^, and promoted the production of excessive ROS and induced ferroptosis in osteosarcoma cells under hypoxia. Liu et al. [52] found that GPX4 was highly expressed in resistant osteosarcoma cells by measuring ferroptosis-related proteins in cisplatin-resistant osteosarcoma cells, and promoting the occurrence of ferroptosis through ferroptosis inducers enhanced the sensitivity of osteosarcoma cells to cisplatin; meanwhile, further studies found that the reduction in GPX4 and the increase in ROS could correspondingly improve the sensitivity of resistant osteosarcoma cells to cisplatin. A study by Lin et al. [53] found that application of *EF24* induces ferroptosis by promoting the formation of ROS. According to Chen et al. [54], KDMA4 promotes cystine transit, inhibiting ferroptosis by demethylating *SLC7A11*. Fu et al. [55] found that nanomedicine accumulates Fe^2+^ and consumes GSH. Lei et al. [56] established an osteosarcoma disease prediction model based on ferroptosis-related genes and achieved good prediction results (Figure 4 and Table 2).

## 6. Intervertebral Disc Degeneration and Ferroptosis 

Lumbar disc herniation (LDH) is one of the common causes of chronic low back and leg pain, mainly after intervertebral disc degeneration (IVDD); then, annulus fibrosus ruptures, nucleus pulposus tissue protrudes into the spinal canal, adjacent nerve roots are compressed, and symptoms such as low back and leg pain and lower limb numbness occur. Zhang et al. identified unique chondrocyte subsets and revealed the involvement of ferroptosis in human intervertebral disc degeneration using single-cell RNA-seq analysis [57]. Zhang et al. [58] concluded that GPX4 methylation is shown to increase oxidative stress and ferroptosis in the nucleus pulposus when homocysteine is present, which is a new factor contributing to IVDD. Under oxidative stress, Lu et al. [59] found that ferroptosis and ferroportin dysfunction contribute to the depletion of nucleus pulposus cells (NPCs) and the development of IVDD. Shan et al. [60] concluded that the Notch pathway might play a role in heme-induced ferroptosis in human nucleus pulposus cells (HNPCs). Yang et al. [61] demonstrated the effects of ferrostatin-1, deferoxamine, and *RSL3* on annulus fibrosus cells (AFCs) and nucleus pulposus cells (NPCs) treated with tert-butyl hydroperoxide (TBHP). Li et al. [62] revealed that ATF3 has the potential to be used as a promising therapeutic target against IVDD. Yu et al. [63] found that bone marrow mesenchymal stem cells-extracellular vesicles (BMSC-EV)-loaded circ_0072464 inhibited NPC ferroptosis to relieve IVDD via upregulation of miR-431-mediated NRF2, indicating a therapeutic target for IVDD (Table 3).

## 7. Spinal Cord Injury and Ferroptosis

Spinal cord injury (SCI) refers to the spinal cord injury caused by external direct or indirect factors, and various motor, sensory, and sphincter dysfunction, abnormal muscle tone, and pathological reflexes occur in the corresponding segments of the injury. The extent and clinical presentation of a spinal cord injury depends on the location and nature of the primary injury. Spinal cord injuries can be divided into primary and secondary spinal cord injuries. The former refers to the injuries caused by external forces acting directly or indirectly on the spinal cord. In the latter, spinal cord damage is further aggravated by spinal cord edema caused by external forces, hematoma formed by intraspinal small vessel bleeding, and compression fractures; the latter is the result of many biochemical changes in the spinal cord caused by primary trauma, including blood–brain barrier function problems, bleeding, partial ischemia, inflammatory response, Ca^2+^ spillage, electrolyte imbalance, apoptosis, oxidative stress, and others. Secondary injury will cause self-destruction of intact tissues around the spinal cord injury and further deepen the degree of SCI. It can be seen that the key to the repair of a spinal cord injury lies in how to effectively inhibit the damage caused by secondary injury to cells. After a spinal cord injury, iron overload is caused by a massive release of iron from hemoglobin due to bleeding at the injury site, and excessive ROS is produced to cause ferroptosis. It has been confirmed by some scholars that ferroptosis is a major cause of the serious consequences of secondary injuries after SCI, and deferoxamine (DFO) inhibits ferroptosis in SCI rats, promoting recovery of motor function; in addition, they found that ferroptosis is linked to mitochondrial changes, thus confirming that ferroptosis is involved in SCI [64]. Zhang et al. [65] found that an intraperitoneally injected ferroptosis inhibitor (SRS16-86) effectively reduced ferroptosis-related 4-hydroxylnonenal (4HNE) levels while increasing GPX4, xCT, and GSH levels, alleviating the damage after SCI in rats. They concluded that deteroxiamine inhibits ferroptosis, making it a promising therapeutic strategy for spinal cord injuries. Other scholars have found that secondary spinal cord injury can lead to a large decrease in oligodendrocytes [66]; a decrease in GSH and a large increase in iron cause oxidative damage to oligodendrocytes [67]. Fan et al. [24] confirmed that *RSL3* induces ferroptosis in oligodendrocytes and found that liproxstatin-1 is more effective than edaravne (EDA) or deferoxamine (DFO) in inhibiting ferroptosis in oligodendrocytes, revealing the important role of ferroptosis in oligodendrocyte survival. Ge et al. found that through NRF2/HO-1 and GPX4 signaling pathways, zinc inhibited ferroptosis in neurons by degrading oxidative stress products and lipid peroxides [68]. Gong found that in mice with SCI, trehalose inhibits ferroptosis via the NRF2/HO-1 pathway and promotes functional recovery [69] (Figure 5). At present, the clinical research on spinal cord injury mostly starts from a single treatment, and for further investigation, multiple treatment methods can be combined. In addition, the therapeutic effect of traditional Chinese medicine on spinal cord injury is considerable, but it still needs subsequent in-depth study to explore the mechanism of improving spinal cord injury from the direction of ferroptosis. On the other hand, neurogenic bladder is one of the most common complications of a spinal cord injury [70]. Further research is needed to determine whether ferroptosis plays a role in its mechanism and whether it improves neurogenic bladder by inhibiting ferroptosis. 

## 8. Conclusions

Ferroptosis, a novel form of cell death, has become a current research hotspot since it was discovered in 2012. Various studies have confirmed that ferroptosis may play a key role in the development of various musculoskeletal diseases, and intervening in the process of ferroptosis can improve the development of related diseases to varying degrees. The current understanding of ferroptosis is not comprehensive. In terms of molecular mechanisms of ferroptosis, iron overload and ROS are important factors in the ferroptosis signaling pathway, and other underlying molecular mechanisms of this process as well as other mechanisms mediating ROS accumulation in mitochondria in addition to the Fenton response still need further investigation. Because the occurrence of ferroptosis involves many aspects and the mechanisms are complex and diverse, many key problems remain to be solved. However, with the advancement of research and technology, targeting ferroptosis is expected to become an entry point for non-surgical treatment and early intervention in musculoskeletal diseases. 

## Figures and Tables

**Figure 1 jcm-12-02125-f001:**
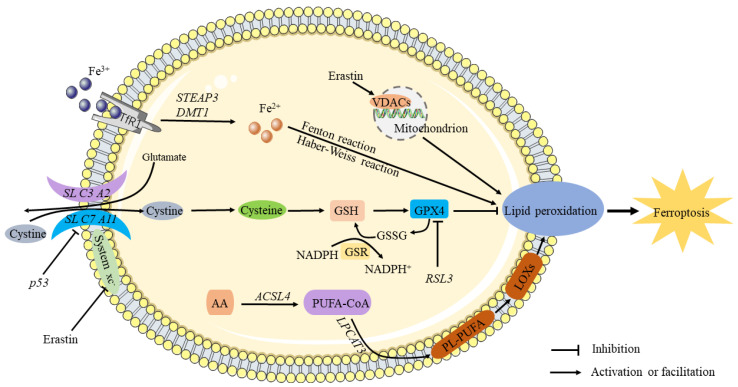
The mechanism of ferroptosis. It is known that circulating iron enters cells by binding to the TfR1 on the cell membrane, where *STEAP3* reduces ferric iron into ferrous iron. The Fenton and Haber–Weiss reactions produce ROS when divalent iron is released by *DMT1* to a labile pool of iron in the cytoplasm, thereby triggering ferroptosis. Extracellular cystine is transported into the cell and converted into cysteine by the sodium-dependent system xc–. Inhibiting system Xc- decreases GSH levels in cells, aggravating ROS accumulation and ultimately causing ferroptosis. Through downregulation of the *SLC7A11* subunit, *P53* inhibits the uptake of cystine by system X, resulting in decreased cystine-dependent glutathione peroxidase activity and cell antioxidant capacity and an increase in lipid ROS, leading to ferroptosis. By antagonizing GPX4, *RSL3* induces ferroptosis. As a result of chain reactions, hydroxyl radicals can directly react with PUFAs in membrane phospholipids to form lipid peroxides, causing ferroptosis.

**Figure 2 jcm-12-02125-f002:**
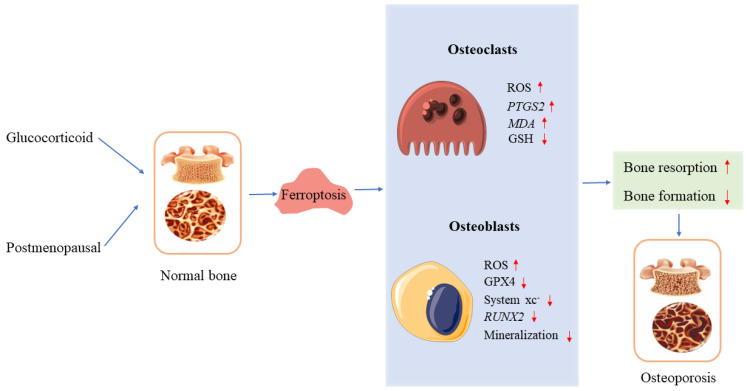
Ferroptosis and osteoporosis. In osteoclasts and osteoblasts, glucocorticoids and postmenopausal hormones induce ferroptosis, leading to osteoporosis. ↑ increase, ↓ decrease.

**Figure 3 jcm-12-02125-f003:**
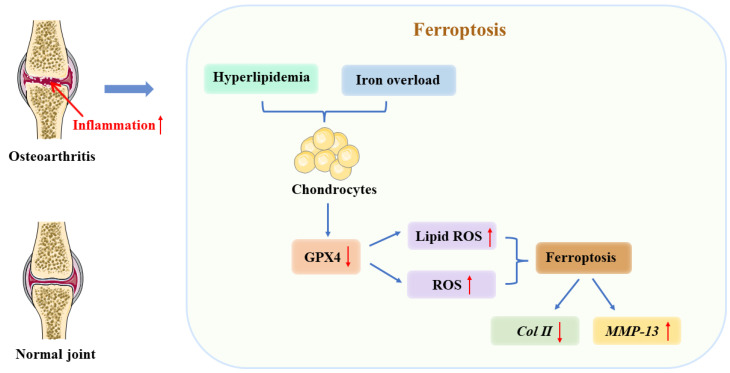
The relationship between ferroptosis and osteoarthritis (OA). As a result of iron overload in cellular environments, hyperlipidemia, or inflammation, chondrocytes express less GPX4. It is these changes that ultimately result in ferroptosis due to the accumulation of reactive oxygen species and lipid peroxides. In turn, ferroptosis contributes to an inflammatory response that results in increased *MMP-13* expression and decreased collagen *II* expression among chondrocytes to accelerate cartilage deterioration. ↑ increase, ↓ decrease.

**Figure 4 jcm-12-02125-f004:**
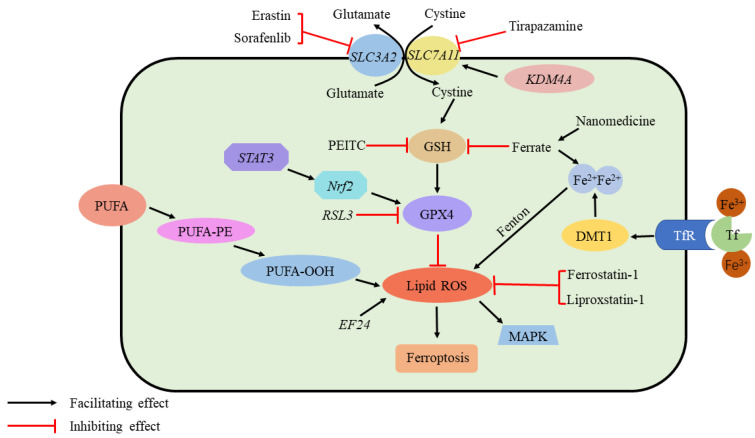
Ferroptosis and osteosarcoma. Ferroptosis requires ROS accumulation. The production of ROS is initiated by the peroxidation of unsaturated fatty acids and iron catalysis. In addition, several elements regulate the GPX4 pathway, which is critical for inhibiting ferroptosis by scavenging ROS: Sorafenib and erastin inhibit cystine transport via System Xc-. As a result of PEITC, GSH level is depleted; ferrate increases iron content as well as depleting GSH. As a result of Tirapazamine, *SLC7A11* is suppressed. *STAT3/Nrf2* facilitates GPX4 production, while *RSL3* inhibits it. While *EF24* promotes ROS formation, Ferrostatin-1 and Liproxstatin-1 suppress it. *KDMA4* promotes cystine transit by demethylating *SLC7A11*. Nanomedicine Accumulates Fe^2+^ and consumes GSH. ROS, reactive oxygen species; PEITC, b-phenethyl isothiocyanate; *STAT3*, signal transducer and activator of transcription 3; *Nrf2*, nuclear factor erythroid 2-related factor 2. *EF24*, 3,5-bis (2-fluorobenzylidine)-4-pyperidone; GPX4, glutathione peroxidase 4; *KDM4A*, lysine demethylase 4A; GSH, glutathione.

**Figure 5 jcm-12-02125-f005:**
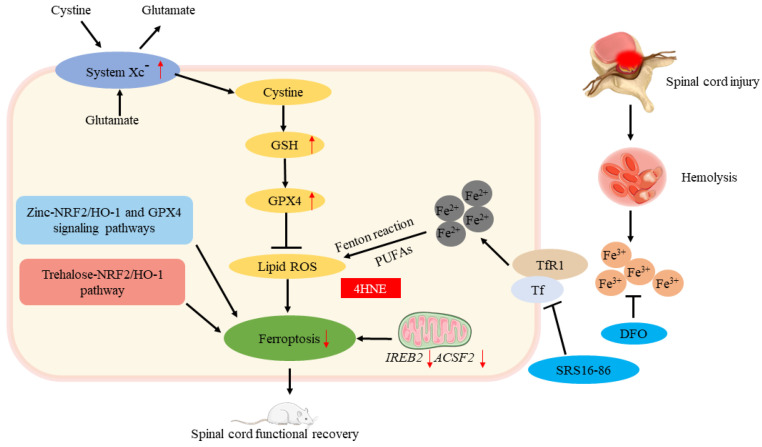
Ferroptosis and spinal cord injury. The roles of ferroptosis in spinal cord injury. ↑ increase, ↓ decrease.

**Table 1 jcm-12-02125-t001:** Characteristics of different programmed cell deaths.

Category	Morphologic Characteristics	Biochemical Profile	Key Factors
Ferroptosis	Mitochondrial atrophy, increased membrane density, reduced or absent mitochondrial cristae, outer membrane rupture	Iron accumulation and lipid peroxidation	*GPX4*, *TFR1*, *SLC7A11*, *Nrf2*, *NCOA4*, *P53*, *HSPB1*, *ACSL4*, *FSP*, etc.
Apoptosis	Karyorrhexis, chromatin condensation, apoptotic body formation and cytoskeletal disassembly, cell size reduction	DNA fragmentation	*caspase*, *Bcl-2*, *Bax*, *p53*, *Fas*, etc.
Autophagy	Bilayer membrane structure (autophagic vacuoles/lysosomes)	Increased lysosomal activity	*ATG5*, *ATG7*, *LC3*, *Beclin-1*, *DRAM3*, *TFEB*, etc.
Pyroptosis	Nuclear condensation, chromatin DNA fragmentation, cell swelling followed by membrane rupture	Activation of caspase	*caspase-1*, *4*, *5*, *11*, *GSDMD*, *Cleaved*, *CASP-3*, *IL-1β*, *IL-18*, etc.
Cuproptosis	Mitochondrial shrinkage, mitochondrial membrane rupture	Cu^2+^ binds directly to the fatty acylated portion of the tricarboxylic acid cycle	*FDX1*, *LIAS*, *LIPT1*, *DLD*, *MTF1*, *GLS*, *CDKN2A*, etc.

*GPX4:* glutathione peroxidase 4; *TFR1*: transferrin receptor 1; *SLC7A11:* solute carrier family 7 member 11; *Nrf2*: nuclear factor E2-related factor 2; *NCOA4*: nuclear receptor coactivator 4; *HSPB1*: heat shock protein B1; *ACSL4*: long-chain fatty acyl-CoA synthase 4; *FSP*: ferroptosis suppressor protein; *Bcl-2*: B-cell lymphoma-2; *Bax*: Bcl-2 associated X protein; *ATG*: autophagy-related gene; *LC3*: microtubule-associated protein 1 light chain 3; *Beclin-1*: Bcl-2 homeodomain protein; *DRAM3*: DNA damage-regulated autophagy modulator3; *TFEB*: transcription factor EB; *RIP*: receptor-interacting protein; *GSDMD*: gasdermin D; Cleaved, *CASP-3*: cystatin-3 activated form; *IL*: interleukin; *FDX1*: ferredoxin 1; *LIAS*: lipoicacid synthetase; *LIPT1*: lipoyltransferase1; *DLD*: dihydrolipoamidedehydrogenase; *MTF1*: metal-regulatory transcription factor 1; *GLS*: glutaminase; *CDKN2A*: cyclin-dependent kinase inhibitor 2A.

**Table 2 jcm-12-02125-t002:** Ferroptosis and osteosarcoma.

Cancer Type	Compound	Target	Effect	Reference
Osteosarcoma	PEITC	Consuming GSH	Inducing ferroptosis	[50]
	Tirapazamine	Suppressing *SLC7A11*	Inducing ferroptosis	[51]
	*STAT3*	Accumulating GPX4	Inhibiting ferroptosis	[52]
	*EF 24*	Accumulating ROS	Inducing ferroptosis	[53]
	*KDM4A*	Accelerating system Xc	Inhibiting ferroptosis	[54]
	Nanomedicine	Accumulating Fe^2+^; consuming GSH	Inducing ferroptosis	[55]

**Table 3 jcm-12-02125-t003:** Mechanism of ferroptosis in IVDD.

Study	Year	Mechanism	Effects of NPC
Zhang et al. [57]	2020	Promotion of methylase expression and the upregulation of the GPX4 methylation	Inducing ferroptosis in NPCs
Lu et al. [59]	2021	FPN downregulation and intercellular iron overload	Inducing ferroptosis in NPCs
Shan et al. [60]	2021	Increased heme catabolism, downregulation of GPX4, and intercellular iron overload, which might be mediated by the Notch pathway	Inducing ferroptosis in NPCs
Yang et al. [61]	2021	NCOA4-mediated ferritinophagy and intercellular iron overload	Inducing ferroptosis in NPCs and AFCs.
Li et al. [62]	2022	Upregulation of ATF3 and ROS products	Inducing ferroptosis in NPCs
Yu et al. [63]	2022	Decreased NRF2 expression and upregulation of ROS products	Inducing ferroptosis in NPCs

IVDD, intervertebral disc degeneration; NPCs, nucleus pulposus cells; FPN, ferroportin; GPX4, glutathione peroxidase 4; NCOA4, nuclear receptor coactivator 4; AFCs, annulus fibrosus cells; ATF3, activation transcription factor 3; ROS, reactive oxygen species; NRF2, nuclear factor-erythroid 2.

## Data Availability

All data analyzed were included in this paper; further requests can be consulted and data can be obtained from the correspondent author.

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
