# Peer review of "The Potential Roles of Ferroptosis in Pathophysiology and Treatment of Musculoskeletal Diseases—Opportunities, Challenges, and Perspectives"

_jcm, 2023, doi:10.3390/jcm12062125_

Round 1
Reviewer 1 Report
Hu, Wang, Liu et al. aimed to perform a review of the current knowledge on the role of ferroptosis in common musculoskeletal disorders, such as osteoarthritis, osteoporosis, osteosarcoma, intervertebral disc degeneration and spine injuries. Since ferroptosis is a relatively recently identified form of cell death, distinct from apoptosis, autophagy, pyroptosis, and cuproptosis, it provides a new potential for clinical diagnosis and treatment of musculoskeletal diseases. The topic of this review is of high interest to the readers, however in my opinion the manuscript needs major revision.
Please find my suggestions below:
1. The English language needs extensive editing. The authors should revise their manuscript carefully to avoid sentences such as: “Both iron overload and lipid peroxide accumulation contribute to the destruction of musculoskeletal diseases, such as osteoporosis, osteoarthritis, osteosarcoma, intervertebral disc degeneration, spinal cord injury, ultimately causing these diseases to occur, develop and progress.” This sentence should be completely rephrased, as iron overload and lipid peroxide accumulation do not cause the destruction of the disease, but rather musculoskeletal tissues and this then leads to the onset of the musculoskeletal diseases that the authors reviewed.
2. The last two sentences in Introduction should be combined or rephrased.
3. The structure should be improved. The chapter 3.2. entitled “Ferroptosis with Osteoclasts and Osteoblasts” does not fit to the concept of reviewing ferroptosis in musculoskeletal diseases and should be rather part of the Ferroptosis and Osteoporosis.
4. Please make sure to write the markers that were measures on gene level in italics, both in text and figures.
5. Please improve the resolution of the Figures. Further make sure that the words are written in capital, i.e. Chondrocytes in Figure 3.
6. Could the authors please provide the criteria for selecting the papers to be included in this review, i.e. how did they perform the database search?
Author Response
We would like to thank the reviewers for thoroughly reviewing our manuscript and making many thoughtful comments. We were very pleased to see that our reviewers recognized the potential significance of our work. We humbly made some changes, described in detail below, and revised the manuscript to address reviewers’ comments. Hopefully it could optimize our paper a little bit. Here are our point-by-point responses: stressed with red text in our reuploaded main manuscript.
Reviewer: 1
1.The English language needs extensive editing. The authors should revise their manuscript carefully to avoid sentences such as: “Both iron overload and lipid peroxide accumulation contribute to the destruction of musculoskeletal diseases, such as osteoporosis, osteoarthritis, osteosarcoma, intervertebral disc degeneration, spinal cord injury, ultimately causing these diseases to occur, develop and progress.” This sentence should be completely rephrased, as iron overload and lipid peroxide accumulation do not cause the destruction of the disease, but rather musculoskeletal tissues and this then leads to the onset of the musculoskeletal diseases that the authors reviewed.
We apologize for the poor language of our manuscript. We worked on the manuscript for a long time and the repeated addition and removal of sentences and sections obviously led to poor readability. We have now worked on both language and readability and have also involved native English speakers for language corrections. We marked the corrections with red texts in our paper. We really hope that the flow and language level have been substantially improved. “Both iron overload and lipid peroxide accumulation contribute to the destruction of musculoskeletal diseases, such as osteoporosis, osteoarthritis, osteosarcoma, intervertebral disc degeneration, spinal cord injury, ultimately causing these diseases to occur, develop and progress.” Corrected as “Both iron overload and lipid peroxide accumulation contribute to the occurrence, development, and treatment of musculoskeletal diseases, such as osteoporosis, osteoarthritis, osteosarcoma, intervertebral disc degeneration, spinal cord injury.” In page 1, line 23-25.
2.The last two sentences in Introduction should be combined or rephrased.
These two sentences have been rephrased as “For better understanding the potential roles ferroptosis may play in pathophysiology and treatment of common musculoskeletal disorders, this article briefly reviewed the relationship and possible mechanisms.” In page2, line 51-54.
3.The structure should be improved. The chapter 3.2. entitled “Ferroptosis with Osteoclasts and Osteoblasts” does not fit to the concept of reviewing ferroptosis in musculoskeletal diseases and should be rather part of the Ferroptosis and Osteoporosis.
We thank our reviewer for pointing out this, we have agreed with our reviewer and combined them into 4. Ferroptosis and Osteoporosis, which is indeed more suitable. We further divided part 4 into 3 parts which are 4.1. Iron Overload; 4.2 ROS; 4.3. Possible Mechanisms in Osteoclasts and Osteoblasts. Page 4-5, line 145-194.
4.Please make sure to write the markers that were measures on gene level in italics, both in text and figures.
Thanks a lot for letting us know this, we made changes accordingly. All gene level markers were changed in italics. (Marked with red in main-manuscript, and Figures were also changed accordingly).
5.Please improve the resolution of the Figures. Further make sure that the words are written in capital, i.e. Chondrocytes in Figure 3.
We highly appreciated your rigour in revising our paper. We have uploaded the PDF version of all figures in supplementary, and “chondrocytes” has been corrected as “Chondrocytes” in Figure 3.
6.Could the authors please provide the criteria for selecting the papers to be included in this review, i.e. how did they perform the database search?
Our reviewer checked our paper in an erudite way, we highly agreed with this comment, and it is really necessary to provide the criteria for selecting the papers. Therefore, we have added this as 2. Materials and Methods. In page 2, line 55-65.
Demonstrated as “In our study, published peer-reviewed journal papers in English were included; PubMed, Google Scholar, and Web of Science databases were searched from inception until October 2022. Keywords used for searching included ferroptosis, and osteoporosis, or osteoarthritis, or osteosarcoma, or intervertebral disc degeneration, or spinal cord injury. Following that, manual searches of the reference lists of the studies found in the databases were conducted. Meta-analyses and review articles were excluded as research sources. All articles (title and abstract) were screened by three authors using the predefined eligibility criteria (Original studies focusing on ferroptosis and the mentioned musculoskeletal diseases). If two of the three authors agreed that the manuscript met the eligibility criteria, it was considered suitable for further analysis (primary literature).”
Reviewer 2 Report
In the manuscript titled “The potential Roles of Ferroptosis in Pathophysiology and Treatment of Musculoskeletal Diseases-Opportunities, Challenges and Perspectives”, the authors reviewed the association and possible pathophysiology between ferroptosis and several musculoskeletal diseases. It is interesting to focus on ferroptosis and musculoskeletal diseases as some diseases are related to iron and lipid overload which can affect ferroptosis and the reviewer consider the authors are appropriate to review this topic since they have already published the paper in which they analyzed ferroptosis associated biomarkers for postmenopausal osteoporosis (PMID 36105394).In order to clear the rationale of this review, the authors need to describe in detail why ferroptosis can affect osteoporosis. In the manuscript, the authors referred #33 and #34 in which the clinical data suggested low level of iron could exacerbate osteoporosis. As the authors explained in introduction, ferroptosis is associated with the iron overload. From these descriptions, the reviewer and readers can understand that the iron overload and ferroptosis will improve osteoporosis. On the other hand, the authors mentioned that the iron inhibited osteogenic differentiation of MSCs as referring #41 and illustrated ferroptosis can lead osteoporosis as shown in figure 2. Moreover, the authors explained T2DM has been associated with osteoporosis, while osteoblasts exposed to high glucose could inhibit ferroptosis by overexpression of FtMt. These descriptions seem that ferroptosis inhibition cause osteoporosis and to be inconsistent. The authors need to explain this paradox more closely.
The authors referred #27 and #28 to show the evidence of iron overload is a risk factor for osteoporosis. However, these references are review articles. It is not preferred to refer the review articles in a review manuscript. The authors need to change these references to original articles.
In line 24-25 and50-52, these two sentences should be one sentence to make sense.
Author Response
Reviewer: 2
1.In the manuscript titled “The potential Roles of Ferroptosis in Pathophysiology and Treatment of Musculoskeletal Diseases-Opportunities, Challenges and Perspectives”, the authors reviewed the association and possible pathophysiology between ferroptosis and several musculoskeletal diseases. It is interesting to focus on ferroptosis and musculoskeletal diseases as some diseases are related to iron and lipid overload which can affect ferroptosis and the reviewer consider the authors are appropriate to review this topic since they have already published the paper in which they analyzed ferroptosis associated biomarkers for postmenopausal osteoporosis (PMID 36105394).In order to clear the rationale of this review, the authors need to describe in detail why ferroptosis can affect osteoporosis. In the manuscript, the authors referred #33 and #34 in which the clinical data suggested low level of iron could exacerbate osteoporosis. As the authors explained in introduction, ferroptosis is associated with the iron overload. From these descriptions, the reviewer and readers can understand that the iron overload and ferroptosis will improve osteoporosis. On the other hand, the authors mentioned that the iron inhibited osteogenic differentiation of MSCs as referring #41 and illustrated ferroptosis can lead osteoporosis as shown in figure 2. Moreover, the authors explained T2DM has been associated with osteoporosis, while osteoblasts exposed to high glucose could inhibit ferroptosis by overexpression of FtMt. These descriptions seem that ferroptosis inhibition cause osteoporosis and to be inconsistent. The authors need to explain this paradox more closely.
Our reviewer has analyzed our paper in an erudite and rigorous way, we highly appreciated our reviewer for pointing out these ambiguous descriptions. In order to clear the ambiguity and rationale of this review, we further divided part 4 into 3 parts, which are 4.1. Iron Overload; 4.2 ROS; 4.3. Possible Mechanisms in Osteoclasts and Osteoblasts. Page 4-5, line 145-194.
References 30-34 have been changed as the following. Page 12, line 443-452.
- Jeney, V. Clinical impact and cellular mechanisms of iron overload-associated bone loss. Front Pharmacol 2017, 8, 77.
- Messer, J.G.; Kilbarger, A.K.; Erikson, K.M.; Kipp, D.E. Iron overload alters iron-regulatory genes and proteins, down-regulates osteoblastic phenotype, and is associated with apoptosis in fetal rat calvaria cultures. Bone 2009, 45, 972-979.
- Jiang, Z.; Wang, H.; Qi, G.; Jiang, C.; Chen, K.; Yan, Z. Iron overload-induced ferroptosis of osteoblasts inhibits osteogenesis and promotes osteoporosis: An in vitro and in vivo study. IUBMB Life 2022, 74, 1052-1069.
- Tsay, J.; Yang, Z.; Ross, F.P.; Cunningham-Rundles, S.; Lin, H.; Coleman, R.; Mayer-Kuckuk, P.; Doty, S.B.; Grady, R.W.; Giardina, P.J., et al. Bone loss caused by iron overload in a murine model: Importance of oxidative stress. Blood 2010, 116, 2582-2589.
- Zhang, J.; Zhao, H.; Yao, G.; Qiao, P.; Li, L.; Wu, S. Therapeutic potential of iron chelators on osteoporosis and their cellular mechanisms. Biomed Pharmacother 2021, 137, 111380.
2.The authors referred #27 and #28 to show the evidence of iron overload is a risk factor for osteoporosis. However, these references are review articles. It is not preferred to refer the review articles in a review manuscript. The authors need to change these references to original articles.
We highly appreciated this valuable comment, we have changed reference 27, 28 to original articles. In page 12, line 437-440.
- Xia, D.; Wu, J.; Xing, M.; Wang, Y.; Zhang, H.; Xia, Y.; Zhou, P.; Xu, S. Iron overload threatens the growth of osteoblast cells via inhibiting the pi3k/akt/foxo3a/dusp14 signaling pathway. J Cell Physiol 2019, 234, 15668-15677.
- Sun, X.; Xia, T.; Zhang, S.; Zhang, J.; Xu, L.; Han, T.; Xin, H. Hops extract and xanthohumol ameliorate bone loss induced by iron overload via activating akt/gsk3β/nrf2 pathway. J Bone Miner Metab 2022, 40, 375-388.
3.In line 24-25 and50-52, these two sentences should be one sentence to make sense.
We appreciated your valuable comments, these sentences are really ambiguous, we have changed as “For better understanding the potential roles ferroptosis may play in pathophysiology and treatment of common musculoskeletal disorders, this article briefly reviewed the relationship and possible mechanisms.” Page 1, line25-28; page 2, line 51-54.
We thank our reviewers again for correcting our paper, we have gotten our paper revised. And we have re-compile our paper. Thanks again for your valuable comments.
Yours sincerely
All authors
2023-2-15
Round 2
Reviewer 1 Report
In my opinion the authors have succesfully revised the manuscript. I have no further comments.
Author Response
We highly appreciated your help and valuable comments in revising our paper.
Yours
Sincerely
All authors
Reviewer 2 Report
In the revised manuscript, the authors have addressed the reviewers’ comments correctly, so that the manuscript has been improved. This manuscript will be considered to be accepted after some minor revisions mentioned as follows.
The authors referred review articles as #30, #34, and #38. However, these references should be original articles because they should be research sources.
In the revised manuscript, the authors excluded the explanation about the relationship between diabetes mellitus and ferroptosis. Therefore, the authors need to modify the figure 2 and associated figure legend in which the description of “Diabetic” and “diabetes mellitus” should be removed.
In line 192, “Iron overload” should be “iron overload”.
Author Response
We would like to thank the reviewers again for thoroughly reviewing our manuscript. Here are our point-by-point responses: stressed with red text in our reuploaded main manuscript.
1.The authors referred review articles as #30, #34, and #38. However, these references should be original articles because they should be research sources.
Thank you for your rigorous reviewing, we agreed with our reviewer that original articles would be much more appropriate and convincing, therefore, we have changed reference 30,34,38 to original articles as follow. Page12, line439-440;448-449; page13, line458-459.
- Ge, W.; Jie, J.; Yao, J.; Li, W.; Cheng, Y.; Lu, W. Advanced glycation end products promote osteoporosis by inducing ferroptosis in osteoblasts. Mol Med Rep 2022, 25.
- Zhang, J.; Zheng, L.; Wang, Z.; Pei, H.; Hu, W.; Nie, J.; Shang, P.; Li, B.; Hei, T.K.; Zhou, G. Lowering iron level protects against bone loss in focally irradiated and contralateral femurs through distinct mechanisms. Bone 2019, 120, 50-60.
- Ni, S.; Yuan, Y.; Qian, Z.; Zhong, Z.; Lv, T.; Kuang, Y.; Yu, B. Hypoxia inhibits rankl-induced ferritinophagy and protects osteoclasts from ferroptosis. Free Radic Biol Med 2021, 169, 271-282.
2.In the revised manuscript, the authors excluded the explanation about the relationship between diabetes mellitus and ferroptosis. Therefore, the authors need to modify the figure 2 and associated figure legend in which the description of “Diabetic” and “diabetes mellitus” should be removed.
Thank you for your valuable comments, we noticed that after reediting this part. we did exclude the explanation about the relationship between diabetes mellitus and ferroptosis. We agreed with our reviewer, and we have made correspondent changes in our figure 2 and figure legends. Page5, line191-194.
3.In line 192, “Iron overload” should be “iron overload”.
“Iron overload” had been changed into “iron overload”. Page5, line189.
We thank our reviewers again for correcting our paper, thanks a lot for your valuable comments.
Yours sincerely
All authors